# Spatial and Seasonal Characteristics of Air Pollution Spillover in China

Baocheng Yu [1], Wei Fang [2,*], Shupei Huang [2], Siyao Liu [2], Yajie Qi [2] and Xiaodan Han [2]

[1] School of Ocean Sciences, China University of Geosciences, Beijing 100083, China; m17857094277@163.com
[2] School of Economics and Management, China University of Geosciences, Beijing 100083, China; hspburn@163.com (S.H.); liusiyao0929@163.com (S.L.); 3007180009@cugb.edu.cn (Y.Q.); hanxiaodan@cugb.edu.cn (X.H.)
[*] Correspondence: davifang@163.com

**Abstract:** Air pollution spillover can cause air pollution to negatively affect neighboring regions. The structure of air pollution spillover varies with changes in season and space. Researching the spatial and seasonal characteristics of air pollution spillover is beneficial for determining air pollution prevention and control policies. First, this paper uses the GARCH-BEKK model to correlate the air pollution spillover among cities. Second, a complex network is constructed, and cities that have stronger spillover correlations are grouped into the same region. Finally, motifs are analyzed regarding the spillover relationships among regions. This paper also compares the structure of air pollution spillover during various seasons. This study determines that every season has a core region where the air pollution spillover exits the region. The core region in the spring is western East China, in the summer it is northern East China, in the autumn it is northern East China, and in the winter it is northern North China. These regions interact with most other regions. Furthermore, in spring and winter, the phenomena of air pollution spillover between regions are stronger than those in summer and autumn. We can weaken the air pollution spillover by controlling the air pollution in core regions.

**Keywords:** air pollution; spillover; complex network; motif; season

## 1. Introduction

With the pace of urbanization in China, air pollution is no longer isolated to individual cities; it can also influence other regions' air quality. This phenomenon has been classified as air pollution spillover [1]. Because China has the characteristics of a large land surface, prominent seasonal change, and unbalanced industrial development, Chinese air pollution shows a very significant seasonal fluctuation [2,3]. Air pollution spillover can have a negative influence on other neighboring regions [4,5], such as increasing the cancer risk of cities' residents, increasing the number of bacterial colonies in the air, etc. [6,7]. Controlling air pollution spillover could reduce the cost of air pollution treatment [8]. Thus, the Chinese government has paid high attention to the phenomenon of air pollution spillover [9]. Therefore, a better comprehension of the spatial and seasonal characteristics of air pollution spillover structures in China is an urgent need.

Numerous studies have researched the structure of Chinese air pollution spillover. Most studies focus on two aspects: spatial and seasonal factors. From a spatial perspective, air pollution spillover has the potential to influence distant regions. This distance scale includes intercity spillover [10] and spillover between cities [7]. Wu et al. used the GARCH-BEKK model to analyze the air pollution spillover pattern among eight cities in Taiwan Province [11]. Cheng et al. analyzed the air pollution spillover among Yangtze River Delta cities and showed that the phenomenon of air pollution spillover was strong in these cities [12]. Additionally, Liao et al. analyzed the air pollution spillover among cities in Guangdong Province and showed that the phenomenon of air pollution spillover

in Guangdong was a serious concern [13]; spillover among large regions has also been analyzed [14]. Luyu Chang et al. divided China into South China, North China, the Sichuan Basin, etc., and analyzed the spillover of air pollution among these regions [15]. Sun et al. used the WRF–CAMx–PSAT model to analyze air pollution spillover among regions in the middle and lower reaches of the Yangtze River. The results showed that air pollution spillover had seasonal characteristics and that its influence can reach distant regions [16]. He et al. analyzed the air pollution spillover from 2003 to 2018 in central China and showed that the structure of air pollution spillover had clear seasonal characteristics [17]. Chang et al. used the HP index to analyze air pollution spillover in the long term and showed that this phenomenon became more pronounced during the 1990s, when China began to develop [15], and involved spillover among countries [18]. Considering seasonal factors, the structure of air pollution spillover has been shown to change between seasons. Chen et al. demonstrated that the monsoon influenced the structure of air pollution spillover. During the monsoon season, the phenomenon of air pollution spillover was also prominent [19]. Zhou et al. showed that the characteristics of air pollution in Chinese cities were influenced by seasons [20]. Gong et al. showed that the air pollution spillover among Chinese cities was stronger in the winter than in the summer [21]. These papers used different methods and analyzed the phenomenon of air pollution spillover from two perspectives: spatial and seasonal.

However, to our knowledge, most previous studies have mainly focused on air pollution spillover among only a few regions, and there is a lack of data on this phenomenon throughout China. Moreover, traditional air pollution models do not consider the uncertainty of air pollution spillover [22,23], which may have caused deviations in the determination of potential sources of air pollution. Therefore, considering the uncertainty of air pollution spillover, the present paper uses a complex network to describe the structure of air pollution spillover [24,25]. To study air pollution spillover throughout China, it is necessary to use a motif algorithm to further analyze the meso structure of air pollution spillover between regions. A motif is a form of algorithm that is adapted to analyze a time series of spillover among large areas and time scales [26,27], and it is the basic structure of a complex network [28]. Based on the complex network, to some extent, the motif algorithm can increase the accuracy of findings regarding the sources of air pollution spillover. In addition, because the structure of air pollution spillover changes with the seasons, this paper also investigates these changes. The results of this investigation may help local governments introduce more targeted policies to prevent the phenomenon of air pollution spillover.

This paper analyzes the spatial structure characteristics of air pollution spillover in Chinese cities and the influence of seasons on the spatial characteristics to provide a basis for the coordinated control of air pollution in China. First, this paper uses the GARCH-BEKK model to extract the air pollution spillover relationship of 366 cities in China and constructed a complex network. Second, the cities with strong spillover relationships are divided into the same region, and the motif is used to analyze the air pollution spillover structure among regions. Finally, this paper compares air pollution spillover during different seasons.

## 2. Materials and Methods

### 2.1. Data

This paper's data source is the official website of the Ministry of Ecology and Environment of the People's Republic of China (https://datacenter.mee.gov.cn/websjzx/queryIndex.vm; accessed on 1 June 2020). The data range is from 01.03.2019–29.02.2020 throughout the four seasons. The main air pollutants include $O_2$, CO, $PM_{2.5}$, $PM_{10}$, $SO_2$ and $O_3$ [29,30]. These pollutants are monitored by the station inside each city. The detection process follows "Ambient air quality standards" (GB 3095–2012) and "Technical specifications for environmental air quality assessment" (HJ 663–2013). The air quality index

(AQI) is used to reflect air pollutants in a comprehensive way [31,32]. Data are selected at three-hour intervals.

*2.2. Method*

2.2.1. Step 1: Extracting the Relationship of Air Pollution Spillover among Cities Based on the GARCH-BEKK Model

This study uses the GARCH-BEKK model to analyze the air pollution relationship between 366 Chinese cities. The GARCH-BEKK model can be used in the research of a time series of spillover. By using this model, we can analyze time series more flexibly, as the model is able to analyze all types of time series with various lengths, smooth or intermittent, and constant or variable. The model has lower dimensions and moderate calculations in comparison to other models. Binary GARCH-BEKK models can estimate the influence of air pollution spillover between cities, and there is no heteroscedasticity of linear equations in the model [33]. The GARCH-BEKK model often used to analyze the synchronization between time series. Taking this paper as an example: if the AQI in city A rises, then AQI in city B also rises, and the AQI in these two cities have stronger synchronization and have a larger GARCH coefficient. For this study, previous research is used to test the data's ARCH effect. If the *p*-value is larger than 0.01, then the data has an ARCH effect, and we can perform further calculations. If not, we omit the city data. If the distance between the cities is greater than 200 km, we assume that there is no air pollution relationship among the cities [34]. The conditional variance equation of GARCH-BEKK is calculated in the following Equations (1) and (2):

$$H(t) = C'C + A'\varepsilon_{t-1}\varepsilon'_{t-1}A + B'H_{t-1}B \tag{1}$$

$$Ht = \begin{bmatrix} c_{11} & 0 \\ c_{21} & c_{22} \end{bmatrix}' \begin{bmatrix} c_{11} & 0 \\ c_{21} & c_{22} \end{bmatrix} + \begin{bmatrix} a_{11} & a_{12} \\ a_{21} & a_{22} \end{bmatrix}' \begin{bmatrix} \varepsilon^2_{1,t-1} & \varepsilon_{1,t-1}\varepsilon_{2,t-1} \\ \varepsilon_{1,t-1}\varepsilon_{2,t-1} & \varepsilon^2_{2,t-1} \end{bmatrix} \begin{bmatrix} a_{11} & a_{12} \\ a_{21} & a_{22} \end{bmatrix} + \ldots +$$
$$\begin{bmatrix} b_{11} & b_{12} \\ b_{21} & b_{22} \end{bmatrix}' \begin{bmatrix} h_{11,t-1} & h_{12,t\_!} \\ h_{21,t-1} & h_{22,t-1} \end{bmatrix} \begin{bmatrix} b_{11} & b_{12} \\ b_{21} & b_{22} \end{bmatrix} \tag{2}$$

where (t) in the equation represents the conditional variance equation, which is a $2 \times 2$ matrix, *C* is an undertriangular matrix, *A* is a $2 \times 2$ matrix that is the coefficient of ARCH, which represents the contribution of the q-order residual to the conditional variance, and *B* is a $2 \times 2$ coefficient matrix that represents the degree of correlation between the current conditional variance and the past conditional variance.

In Equation (2), $H_t = \begin{bmatrix} h_{11} & h_{12} \\ h_{21} & h_{22} \end{bmatrix}$ can be decoupled as the following Equations (3) and (4):

$$h_{11,t} = c^2_{11} + c^2_{21} + a^2_{11}\varepsilon^2_{1,t-1} + 2a_{11}a_{12}\varepsilon_{1,t-1}\varepsilon_{2,t-1} + a^2_{21}\varepsilon^2_{2,t-1} + b^2_{11}h_{11,t-1} + \ldots + 2b_{11}b_{21}h_{12,t-1} + b^2_{21}h_{22,t-1} \tag{3}$$

$$h_{22,t} = c^2_{11} + a^2_{12}\varepsilon^2_{1,t-1} + 2a_{11}a_{22}\varepsilon_{1,t-1}\varepsilon_{2,t-1} + a^2_{22}\varepsilon^2_{2,t-1} + b^2_{12}h_{11,t-1} + \ldots + 2b_{12}b_{22}h_{12,t-1} + b^2_{22}h_{22,t-1} \tag{4}$$

The parameter matrices A and B in Equations (3)'s and (4)'s elements in diagonal ($a$12, $a$21 and $b$12, $b$21) show how air pollution spills over between cities. To analyze the air pollution spillover between cities, we formulate (5) and (6) as follows:

$$H_0 : a_{12} = a_{21} = b_{12} = b_{21} = 0 \ (no \ spillover \ effect) \tag{5}$$

$$H_1 : a_{12} \neq a_{21} \neq b_{12} \neq b_{21} \neq 0 \ (spillover \ effect), \tag{6}$$

Thus, (5) is the null hypothesis, and (6) is the alternative hypothesis. This result can be used as evidence of air pollution spillover between cities.

2.2.2. Step 2: Dividing the Regions of Air Pollution Spillover Based on a Complex Network

To analyze the division of regions, this paper constructs a complex network of air pollution spillover between cities. This complex network uses cities as nodes and the spillover relationship between two cities closer than 200 km as edges. We use this complex network to research the topology of air pollution between cities. The relationship of nodes and sides is determined by the following Equation (7):

$$G = (N, E) \tag{7}$$

where G represents the air pollution spillover complex network, E represents the edges of the network, and the nodes are represented by *i* and *j*. If there is a relationship between *i* and *j*, we use *ei* and *ej* as the edges between nodes. The equation of the air pollution complex network is as follows (8):

$$E = \begin{bmatrix} e1,1 & \cdots & e1,j \\ \vdots & \ddots & \vdots \\ ei,1 & \cdots & ei,j \end{bmatrix} \tag{8}$$

In this paper, 366 Chinese cities are used as nodes. The coefficient absolute value of GARCH-BEKK spillover is the side's weight. The direction of spillover is the side's direction, as described by the air pollution complex network.

We use the complex network's algorithm of modularity to divide the region. Modularity is a kind of standard that can estimate the division of regions; if the modularity is larger, the network is more strongly divided. The modularity's value is between −1 and 1. The equation of modularity is the following Equation (9) [35]:

$$Q = \frac{1}{2m} \sum_{i,j} \left[ w_{i,j} - \frac{A_i A_j}{2m} \right] \delta(c_i, c_j) \tag{9}$$

In Equation (9), $w_{i,j}$ is the weight of the side between node *i* and node *j*, $A_i = \sum_j w_{i,j}$ is the sum of all sides of *i*'s weight, $A_j = \sum_i w_{j,i}$ is the sum of all sides of *j*'s weight, $m = \frac{1}{2} \sum_{i,j} w_{i,j}$ is the modularity of *i*, and $c_j$ is the modularity of *j*. If node *i* and node *j* are in the same modularity, then $c_i = c_j$, $\delta(c_i, c_j) = 1$; if not, $\delta(c_i, c_j) = 0$. We divide the regions of air pollution such that they can divide the air pollution spillover more synchronously into the same region. This efficiency management can be improved. After dividing the regions, we use ArcGIS to visualize the results.

2.2.3. Step 3: Analyzing the Structure of Air Pollution Spillover among Regions Based on Motifs

The motif is the basic unit of a complex network. The motif is a kind of structure which size between node and modularity in complex network, it is the basic topologic of complex network. A motif is the basic connection method of nodes in complex network. Analyzing motifs can help us better research the model of air pollution spillover between regions. The motifs in the complex network have 13 kinds of 3 nodes. Nodes have 30 kinds of features. All features are shown in Figure 1 [36].

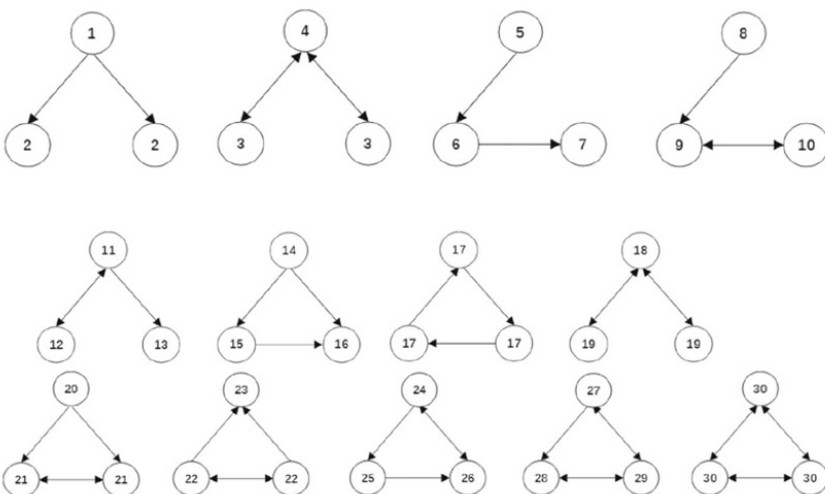

**Figure 1.** Form of 3-node motifs.

In the motif standards, the Z-score represents the ratio of the regular network and the random network. *Nreali* represents the motif in the regular network. *Nrandi* is the motif in the random network. ⟨*Nrandi*⟩ represents the average value. In addition, $\sigma$ *randi* represents the standard deviation. The Z-score Equation (10) is as follows:

$$Zi = \frac{Nreali - \langle Nrandi \rangle}{\sigma randi} \tag{10}$$

Equation (10) calculates the ratio of motifs in the ratio and random network. If the Z-score is positive, the ratio of motifs in the regular network is greater than in the random network and motifs are more important. If the Z-score is negative, the motif is not in the complex network.

The *p*-value shows the frequency of motifs appearing in the complex network. A smaller *p*-value indicates a more important motif. If we remove this motif, we can weaken the spillover of air pollution in regions.

## 3. Results

### 3.1. Regional Division Based on Air Pollution Spillover

Our paper uses a complex network based on the spillover of air pollution that divided the regions and sequentially used ArcGIS to visualize the regions of air pollution, as shown in Figure 2. Red indicates the relationship of air pollution spillover between cities in regions. Darker red indicates that the relationship of cities in regions is stronger. Lighter red indicates that the relationship is weaker. Gray indicates that the average relationship between cities in this region is 0. White signifies that there are no data in this region. Figure 2 shows the spatial distribution of air pollution spillover in different seasons.

In the spring, there are 77 spillover regions. There are 14 regions whose average spillover relationship is not 0. As shown in Figure 2, the regions whose average spillover relationship is not 0 are concentrated between East and North China, as well as between East China and Middle China. In West China, the spillover relationship between cities is very weak. In the spring, there is a strong abnormal northeast wind through the South China Sea and the region of the North Pacific Ocean, which may be the cause of this phenomenon. In the summer, there are 103 spillover regions. There are 14 regions whose average spillover relationship is not 0. From the spatial distribution, the summer regions do not completely border each other. They can be divided into East China and South China, West and North China, and West and South China. The summer characteristics of the spatial distribution of the air pollution spillover relationship from west to the east gradually weaken and then become strong. The regions in the summer are the same as those in the spring, but there are fewer regions whose average relationship is not 0 in the summer;

thus, the ratio of these regions is lower. The cause of this phenomenon is likely from the influences of stronger typhoons, and the South Asian summer monsoon and East Asian summer monsoon cross influences are stronger [37–39]. Autumn has 93 spillover regions. There are 12 regions whose average spillover relationship is not 0. From Figure 2, the difference between autumn, summer, and spring is that in autumn, the average spillover relationship is stronger in the west than in the east. The average spsillover factor is far weaker than in the summer and spring. The cause of this phenomenon is likely from the Northern Hemisphere, and atmospheric migration from west to the east is active [40]. In the winter, there are 79 regions. There are 15 regions whose average spillover relationship is not 0. Winter has the highest number of regions of the seasons. The characteristic in the winter is that the north relationship is stronger than the south relationship. The cause of this phenomenon may be that there is centralized heating in North China during the winter.

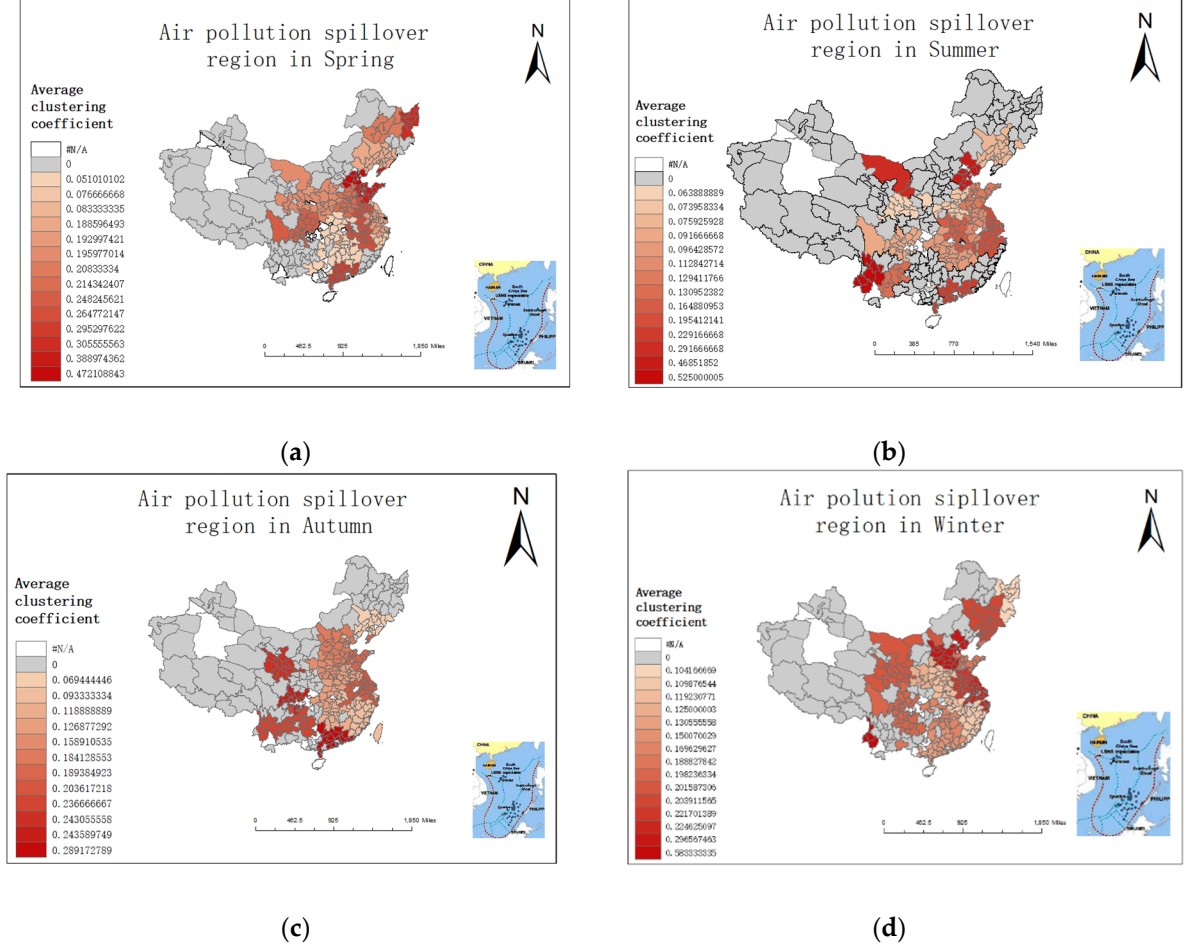

**Figure 2.** Division of air pollution spillover areas over four seasons: (**a**) Air pollution spillover region in Spring, (**b**) Air pollution spillover region in Summer, (**c**) Air pollution spillover region in Autumn, and (**d**) Air pollution spillover region in Winter.

In summary, from a time perspective, the phenomenon of air pollution is stronger in the spring and winter, and the range of spillover is larger. In summer and autumn, the air pollution is weaker, and the range of spillover is smaller. From a spatial perspective, in the spring and winter, air pollution spillover is more common in the middle and east cities. In the summer and autumn, air pollution spillover is observed more in western China.

### 3.2. Air Pollution's Spillover among Regions

This paper uses the number of cities between regions that have a spillover relationship to construct a complex network. The nodes of the complex network are the regions. The color of nodes indicates the strength of the regions' output air pollution. The warmer colors indicate that the output of air pollution is stronger. The cooler colors indicate that the output of air pollution is weaker. The sides of the complex network are the numbers of cities with spillover relationships between regions. Wider sides indicate larger quantities of cities with relationships, and narrow sides mean smaller quantities. Figure 3 shows the complex network constructed by air pollution spillover between regions.

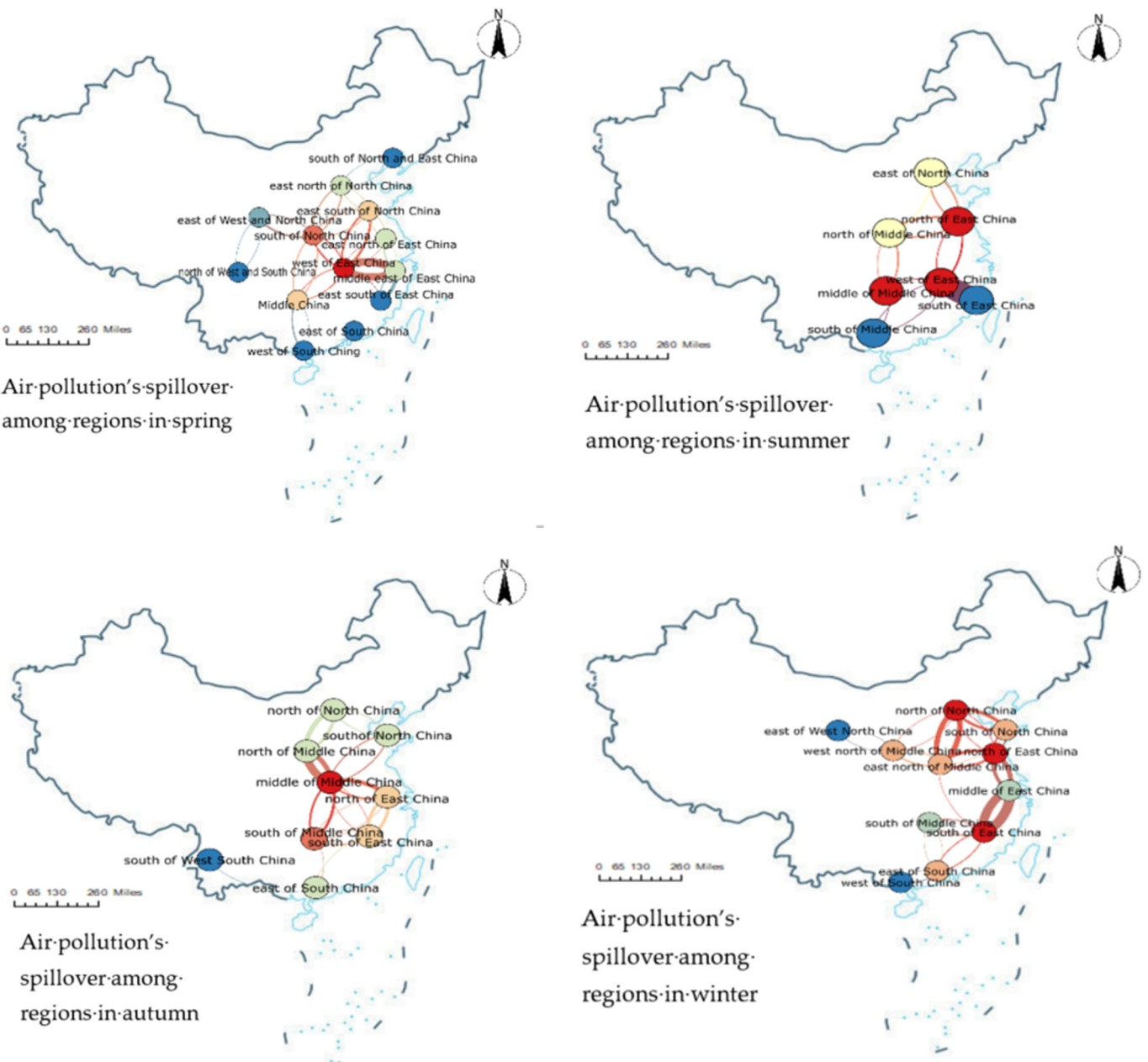

**Figure 3.** Air pollution spillover among regions.

Figure 3 shows that in spring, there are 13 regions with air pollution relationships, which is the most of the four seasons. The regions with strong relationships lie in North China and Middle China. The region with the strongest output of air pollution is the western part of East China. Although cities in eastern China have strong air pollution

spillover relationships, the region does not have relationships with other regions. In the summer, there are seven regions with air pollution spillover relationships. This is the season with the fewest regions with air pollution spillover relationships of the four seasons. The northern part of East China, the western part of East China and the middle of Middle China are the regions that have the strongest spillover relationships. In autumn, nine regions have air pollution spillover relationships. The region that most strongly influenced the other regions is Middle China; other regions have weaker influences. In the winter, there are 11 regions with air pollution relationships. The northern part of North China, the northern part of East China, and the northern part of South China have stronger relationships, and in North China, the air pollution relationship between regions is stronger than in South China.

The air pollution spillover relationship between regions is stronger in the spring and winter and weaker in the summer and autumn. In the summer, spring, and autumn, East China and Middle China have stronger air pollution output into other regions. In the winter, East China, Middle China, and North China have stronger influences on other regions. In the four seasons, the regions with air pollution spillover relationships are concentrated in the Middle China and East China regions. Although in West China, the air pollution spillover relationships in cities are stronger than those in Middle and East China, the air pollution spillover relationship between regions is weaker than that in Middle and East China. The reason perhaps is because cities are dense in East China, with higher development and flat terrain, making it easy to spread air pollutants. West China has complex terrain, and it is difficult for air pollutants to move between regions.

### 3.3. Structure of Air Pollution Spillover among Regions

It is necessary to control air pollution spillover between regions, but this does not mean that every region must be tended to. To analyze the motifs of the air pollution complex network, we identify core regions and central motifs. In this way, we can control air pollution spillover between core regions, destroying the essential construct of the complex network; furthermore, we can weaken the phenomenon of air pollution spillover between regions.

In motifs, if the Z-score is a positive value and undefined, this means that this motif has a significant presence in the complex network. A *p*-value near 0 shows that it is important in the complex network, in contrast to a *p*-value near 1. Tables 1–4 show the construction of the motif in the four seasons. Many previous studies have aimed to describe the form of motifs. We comprehensively describe programs such as Shai S. Shen-Orr (2002), R. Milo (2002), S. Mangan (2003), Ron Milo (2004), and N. Kashtan (2005) [28,41–43]. We correspondingly construct Tables 1–4.

**Table 1.** Spring significant phantom importance evaluation and inclusion region.

| Season | Motif | Important | | Included Regions | | |
|---|---|---|---|---|---|---|
| | Construct | Numbering | Z-Score | *p*-Value | Region 1 | Region 2 | Region 3 |
| Spring |  | #1 Complete connection | 7.7573 | 0 | northeast of North China Middle China west of East China west of East China west of East China | south of North China south of North China north south of East China north south of East China south of North China | southeast of North China west of East China southeast of North China east middle of East China east south of North China |

**Table 1.** *Cont.*

| Season | Motif | | Important | | Included Regions | | |
|---|---|---|---|---|---|---|---|
| | Construct | Numbering | Z-Score | *p*-Value | Region 1 | Region 2 | Region 3 |
| |  | #2 Max output | 0.086047 | 0.452 | Middle China west of East China west of East China west of East China | west of East China southeast of East China southeast of East China southeast of East China | southeast of East China north south of East China south of North China southeast of North China |
| |  | #3 Include core | 6.0967 | 0 | west of East China | southeast of East China | east middle of East China |

**Table 2.** Summer significant phantom importance evaluation and inclusion region.

| Season | Motif | | Important | | Include Regions | | |
|---|---|---|---|---|---|---|---|
| | Construct | Numbering | Z-Score | *p*-Value | Region 1 | Region 2 | Region 3 |
| Summer |  | #3 Include core | 69.254 | 0 | north of Middle China north of Middle China south of Middle China south of Middle China west of East China west of East China | middle of Middle China north of East China west of East China west of East China north of East China north of East China | north of East China west of East China north of East China south of East China south of East China east of North China |
| |  | #4 Uncomplete connection | undefined | 0 | south of Middle China | middle of Middle China | west of East China |
| |  | #5 Notable Core | 27.037 | 0 | north of Middle China | east of North China | west of East China |

**Table 3.** Autumn significant phantom importance evaluation and inclusion region.

| Season | Motif | | Important | | Include Regions | | |
|---|---|---|---|---|---|---|---|
| | Construct | Numbering | Z-Score | *p*-Value | Region 1 | Region 2 | Region 3 |
| Autumn |  | #1 Complete connect | 2.0143 | 0 | north of East China north of East China north of East China south of Middle China | south of East China south of East China middle of Middle China south of East China | middle of Middle China south of Middle China south of Middle China middle of Middle China |
| |  | #6 One node output | 0.7521 | 0 | east of South China | south of Southwest China | south of East China |

**Table 4.** Winter significant phantom importance evaluation and inclusion region.

| Season | Motif | | Important | | Include Regions | | |
|---|---|---|---|---|---|---|---|
| | Construct | Numbering | Z-Score | *p*-Value | Region 1 | Region 2 | Region 3 |
| Winter |  | #1 Complete connect | 5.8694 | 0 | east of South China south of North China northeast of Middle China northeast of Middle China | south of East China north of East China northwest of Middle China north of East China | south of Middle China north of North China north of North China north of North China |
| |  | #7 One node input | 1.7636 | 0 | south of North China | northeast of Middle China | south of East China |

Motif #1 is the construct of complete connection. In this construction, air pollution spillover in three regions can influence each other. Motif #2 is the maximum output construct. Region 3 sends air pollution to regions 1 and 2, and thus, region 3 is the maximum output region. In motif #3, node 1 receives air pollution spillover from regions 1 and 2 and is the notable core of the motif. In motif #4, region 3 receives air pollution spillover from regions 1 and 2. Regions 1 and 2 do not connect. Thus, motif #3 is an incomplete connection construct. In motif #5, regions 1 and 2's air pollution spillovers influence each other, and region 3 sends air pollution spillover influence to regions 1 and 2; thus, region 3 is motif #5's notable core. In motif #6, region 3 sends air pollution spillover to regions 1 and 2. It is the construct of one node output. In motif #7, region 2 sends air pollution spillover to regions 1 and 3; it is the construct of one node received.

From Tables 1 and 4 and Figure 4, motif #1 has the construct of a complete connection. It is the most important during the spring, autumn, and winter. Motif #4, which has an incomplete connection construct, is the most important during the summer. In motif #1, three nodes can transport air pollution spillover influence each other. This type of motif should be the focus of prevention. Motif #4 should be the focus of prevention during the summer.

In the spring, the core of motif #1 is western East China. It can connect with southern North China, northeast East China, middle-east East China, southeast North China, and Middle China; the air pollution spillovers can influence each other. In summer, the core of motif #4 is northern East China. It can connect with northern Middle China, central Middle China, western East China, southern Middle China, southern East China, and eastern North China. In autumn, the core of motif #1 is northern East China. It can connect with southern East China, central Middle China, and southern Middle China. In winter, the core of motif #1 is northern North China. It can connect with eastern South China, southern East China, southern Middle China, southern North China, northern East China, northeast Middle China and northwest Middle China.

Thus, there is a core of air pollution transport in the four seasons. Through the core region, other regions can connect to each other. The different seasons have different core regions. By controlling the air pollution in core regions, we can better prevent air pollution transportation between regions.

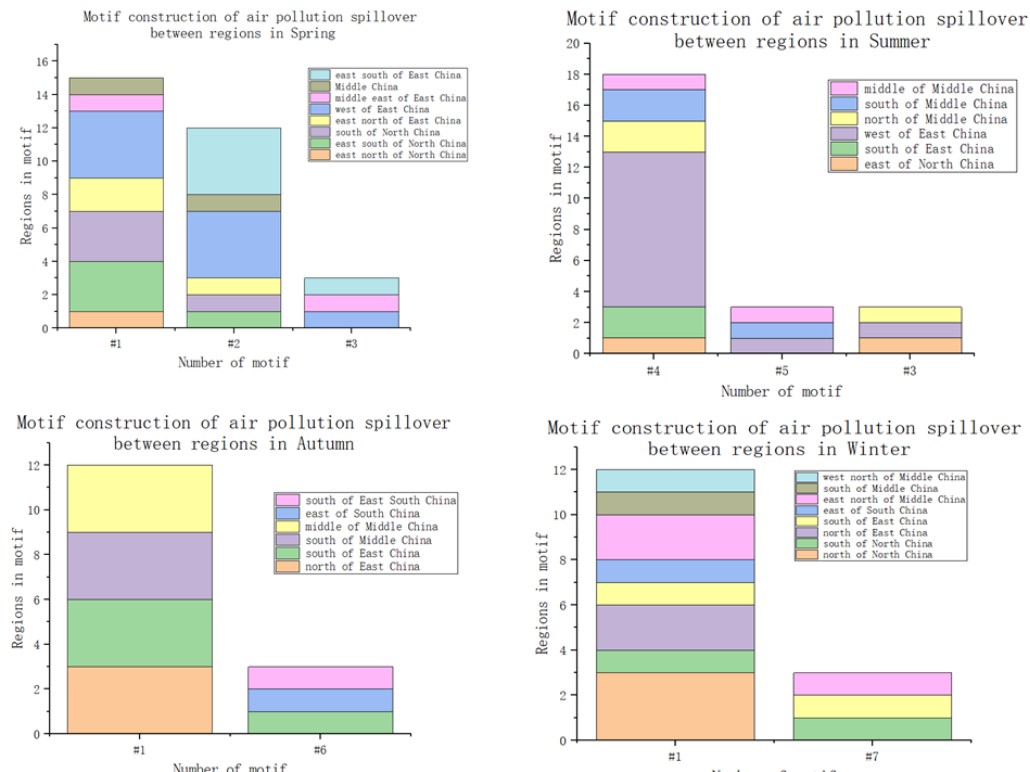

**Figure 4.** Regions included in the significant motif of four seasons.

## 4. Discussion and Conclusions

This paper aims to solve how air pollution spillovers occur between regions in China. Because different seasons have distinct characteristics in China, how air pollution spills over between seasons is a severe problem to address. This paper uses the AQI to characterize air pollution. Then, this paper uses the GARCH-BEKK model to construct an air pollution complex network of 366 Chinese cities. After constructing a complex air pollution network, we use a modularity algorithm to divide the cities that are more synchronous into the same regions. Finally, we use motifs to analyze the structure of air pollution spillover among regions and observe the differences between seasons. Our conclusions are as follows:

- From a seasonal perspective, the phenomenon of air pollution spillover is stronger in spring and winter and weaker in summer and autumn.

The strength of air pollution spillover is different in different seasons. The number of areas that have air pollution spillover connected with other regions is dynamic. In spring, there are 13 regions that have air pollution spillover connected with other areas; in summer, there are 7 regions; in autumn, there are 9 regions; and in winter, there are 11 regions. First, the air pollution spillover among regions is stronger in spring, perhaps caused by the influence of monsoons. Then, the air pollution spillover among summer and winter is weaker. This may be caused by relatively stable atmospheric activity. Finally, the air pollution spillover becomes stronger again in winter. Air pollution influences transport from regions in North China to South China through the regions on the east coast of China. This may be caused by the policy of centralized heating in northern China.

- From a spatial perspective, in general, the regions in North China often transport a negative influence to regions in East China through air pollution spillover.

The air pollution spillover among areas in East China is weaker. They mainly receive a negative influence from regions in northern China and central China. The reason, perhaps, is that limited by the terrain on the east coast of China, the air pollution spillover in these regions is weaker or undetectable. In addition, the areas in southern middle China and

central eastern East China have stronger connections with regions in eastern China. The reason perhaps is that the North China Plain and Huaibei Plain have topographies that are suitable for air pollution spillover. Moreover, the cities in North China have relatively developed industrial levels, and the cities on the east coast of China have relatively developed economic levels. Therefore, the cities' regions in the North China Plain more easily have a negative influence on regions on the east coast of China.

- There is a core region in the air pollution spillover network, and each season has a different core region.

From the results of the analyzed motifs, the four seasons are found to have a center motif, and the center motif has a core region. The core regions are different during the four seasons. The core region in the spring is western East China, in the summer it is northern East China, in the autumn it is northern East China, and in the winter it is northern North China. This core region can transport air pollution to most other regions. This finding shows that when controlling the air pollution spillover, it is not necessary to treat all regions. The most efficient method focuses on air pollution emissions by the core region. If we decoupled the core region from the other regions, the air pollution spillover would be weaker. In addition, because the core region in each season is different, a dynamic management mechanism should be established, and different seasons should be treated differently.

In summary, this paper researches the features of air pollution spillover in all areas in China. This provides a basis for air pollution spillover control on a large scale. However, there is a lack of research on the law of air pollution spillovers between cities on a microscale in this paper, which we plan to resolve in future studies.

**Author Contributions:** Conceptualization, W.F.; Data curation, B.Y., Y.Q. and X.H.; Formal analysis, B.Y. and S.L.; Investigation, B.Y.; Methodology, B.Y. and S.H.; Software, B.Y.; Supervision, W.F.; Writing—original draft, B.Y.; Writing—review & editing, W.F. All authors have read and agreed to the published version of the manuscript.

**Funding:** This research was funded by the National Natural Science Foundation of China, grant number 71991481, 71991480, and 71991485; the Beijing Social Science Foundation, grant number 20GLB016.

**Institutional Review Board Statement:** Not applicable.

**Informed Consent Statement:** Not applicable.

**Data Availability Statement:** Data available in a publicly accessible repository.

**Acknowledgments:** 

**Conflicts of Interest:** The authors declare no conflict of interest.

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
