# Peer review of "Spatial and Seasonal Characteristics of Air Pollution Spillover in China"

_sustainability, doi:10.3390/su132112272_

Round 1
Reviewer 1 Report
The comments are in the paper.

Author Response
Dear Editor and Reviewers:
Thank you for your letter and comments concerning our manuscript entitled “Spatial and seasonal characteristics of air pollution spillover in China” (sustainability-1390246). The comments were all valuable and very helpful for us in revising and improving our paper, and they provided important significant guidance to our research. We have studied the comments carefully and made corrections that we hope meet with your approval. The main corrections in the paper and the response to the reviewers’ comments are below. The revisions are marked in red.
Point 1: Please write as 01.03.2019.-29.02.2020
Response 1: On page 2, ”2019.3.1–2020.2.29” has been changed to “01.03.2019.-29.02.2020”.
Point 2: You need to write how this pollutants were monitored and where the stations were, in the city, near the city...
Response 2: On page 2, we added “These pollutants were monitored by the station inside each city. The detection process followed <Ambient air quality standards> (GB 3095 - 2012) and <Technical specifications for environmental air quality assessment> (HJ 663 - 2013).”
Point 3: The font and font size are different.
Response 3: The font and size have been changed on page 3.
Point 4: checked the number of tables and figures
Response 4: The phrase “fig 3-1” on page 7 has been changed to “Figure 2”; “tables 3-1 to 3-4” on page 9 has been changed to “Table 1 to 4”
Point 5: add literatures
Response 5: On page 7, we refer to refs. [39-41] and ref. [42].
Add literature [39-41] to prove ” The cause of this phenomenon was is likely from the influences of stronger typhoons, and the south Asian summer monsoon and east Asian summer monsoon cross influ-ences were are stronger”
- GU Lianglei, YAO Jimin, HU Zeyong. TYPICAL STRONG AND WEAK SOUTH ASIAN SUMMER MONSOON YEARS AND SEA SURFACE TEMPERATURE OF ARABIAN SEA. Journal of Tropical Meteorology. 2006, 22, 374-379
- Ju Jianhua, Qian Cheng, Cao Jie. The Intraseasonal Oscillation of East Asian Summer Monsoon. Chinese Journal of Atmospheric Sciences. 2005, 29, 187-194.
- Huang Q, Guan YP. Does the Asian monsoon modulate tropical cyclone activity over the South China Sea? CHINESE JOURNAL OF OCEANOLOGY AND LIMNOLOGY.2012, 30, 960-965.
Add literature [42] to prove ” Northern Hemisphere, and atmospheric migration from west to the east is active”
- ZHOU You, GUAN Zhaoyong, ZHANG Qian, YIN Yangyan. Circulation anomalies and their impacts on autumn temper-ature variations over China in associationwith different phase combinations of atmospheric mass migration between lands and oceans and interhemispheric oscillations. ACTA Meteorologica Sinica. 2016, 74,367-379.
Thank you very much for your comments on our article. These comments not only improve the paper but also are of great help to our future scientific research.
The specific modification is in the attachment; please confirm. Thank you very much.
Best regards,
Baocheng Yu

Reviewer 2 Report
The spatial structure characteristics of air pollution spillover in 77 Chinese cities and the influence of seasons on the spatial characteristics to provide a basis 78 for the coordinated control of air pollution were analyzed in this study. I think this paper is a meaningful enough result. However the followings should be revised :
At first, it is not easy to understand the expression and contents of this manuscript. English proof should be needed.
Second, Basically there is no Discussion Section. Even though you have provided the interpretation in Results Section, you should describe the Discussion section.
Third, in Method Section "This paper’s data source was the official website of the Ministry of Ecology and En- 87 vironment of the People's Republic of China (https://datacenter.mee.gov.cn/websjzx/que- 88 ryIndex.vm). The data range is from 2019.3.1–2020.2.29 throughout the four seasons. Cur- 89
rently, the main air pollutants include O2, CO, PM2.5, PM10, SO2 and O3 [28-29]. The air 90 quality index (AQI) was used to reflect air pollutants in a comprehensive way [30-31]. 91 Data were selected at three-hour intervals."
-> why did you make use of 'three-hour intevals'?.
Lastly, You authors would be better explain the algorithms more specifically.
This study would be fine and good results, I think. But need the revision.
Author Response
Dear Editor and Reviewers:
Thank you for your letter and comments concerning our manuscript entitled “Spatial and seasonal characteristics of air pollution spillover in China” (sustainability-1390246). The comments were all valuable and very helpful for us in revising and improving our paper, and they provided important significant guidance to our research. We have studied the comments carefully and made corrections that we hope meet with your approval. The main corrections in the paper and the response to the reviewers’ comments are below. The revisions are marked in red.
Point 1: At first, it is not easy to understand the expression and contents of this manuscript. English proof should be needed.
Response 1: we invited the experts of AJE to help us polished this paper again. The screenshot of polish certificate showed as link 1, please check the link:
Link 1: polish certificate of AJE
https://secure.aje.com/api/certificate/A8A4-3BBF-4042-7970-B342/pdf
Point 2: Second, Basically there is no Discussion Section. Even though you have provided the interpretation in Results Section, you should describe the Discussion section.
Response 2: The discussion and conclusion have been rewrite again. Please check the part 4:
Discussion and conclusion.
- Discussion and conclusion
This paper aims to solve how air pollution spillovers occur between regions in China. Because different seasons have distinct characteristics in China, how air pollution spills over between seasons is a severe problem to address. This paper uses the AQI to characterize air pollution. Then, this paper uses the GARCH-BEKK model to construct an air pollution complex network of 366 Chinese cities. After constructing an air pollution complex network, we use a modularity algorithm to divide the cities that are more synchronous into the same regions. Finally, we use motifs to analyze the structure of air pollution spillover among regions and observe the differences between seasons. Our conclusions are as follows:
- From a seasonal perspective, the phenomenon of air pollution spillover is stronger in spring and winter and weaker in summer and autumn.
The strength of air pollution spillover is different in different seasons. The number of areas that have air pollution spillover connected with other regions is dynamic. In spring, there are 13 regions that have air pollution spillover connected with other areas; in summer, there are 7 regions; in autumn, there are 9 regions; and in winter, there are 11 regions. First, the air pollution spillover among regions is stronger in spring, perhaps caused by the influence of monsoons. Then, the air pollution spillover among summer and winter is weaker. This may be caused by relatively stable atmospheric activity. Finally, the air pollution spillover becomes stronger again in winter. Air pollution influences transport from regions in North China to South China through the regions on the east coast of China. This may be caused by the policy of centralized heating in northern China.
- From a spatial perspective, in general, the regions in North China often transport a negative influence to regions in East China through air pollution spillover.
The air pollution spillover among areas in East China is weaker. They mainly receive a negative influence from regions in northern China and central China. The reason perhaps is that limited by the terrain on the east coast of China, the air pollution spillover in these regions is weaker or undetectable. In addition, the areas in southern middle China and central eastern east of east China have stronger connections with regions in eastern China. The reason perhaps is that the North China Plain and Huaibei Plain have topography that is suitable for air pollution spillover. Moreover, the cities in North China have relatively developed industrial levels, and the cities on the east coast of China have relatively developed economic levels. Therefore, the cities’ regions in the North China Plain more easily have a negative influence on regions on the east coast of China.
- There is a core region in the air pollution spillover network, and each season has a different core region.
From the results of the analyzed motifs, the four seasons are found to have a center motif, and the center motif has a core region. The core regions are different during the four seasons. The core region in the spring is western East China, in the summer it is northern East China, in the autumn it is northern East China, and in the winter it is northern North China. This core region can transport air pollution to most other regions. This finding shows that when controlling the air pollution spillover, it is not necessary to treat all regions. The most efficient method focuses on air pollution emissions by the core region. If we decoupled the core region from the other regions, the air pollution spillover would be weaker. In addition, because the core region in each season is different, a dynamic management mechanism should be established, and different seasons should be treated differently.
In summary, this paper researches the features of air pollution spillover in all areas in China. This provides a basis for air pollution spillover control on a large scale. However, there is a lack of research on the law of air pollution spillovers between cities on a micro scale in this paper, which we plan to resolve in future studies.
Point 3: Third, in Method Section "This paper’s data source was the official website of the Ministry of Ecology and Environment of the People's Republic of China (https://datacenter.mee.gov.cn/websjzx/que- 88 ryIndex.vm). The data range is from 2019.3.1–2020.2.29 throughout the four seasons. Cur- 89
rently, the main air pollutants include O2, CO, PM2.5, PM10, SO2 and O3 [28-29]. The air 90 quality index (AQI) was used to reflect air pollutants in a comprehensive way [30-31]. 91 Data were selected at three-hour intervals."
-> why did you make use of 'three-hour intevals'?.
Response 3: This paper researched the seasonal feature of air pollution spillover. Therefore, used daily data was enough. However, to reflect the AQI’s change in each day, we chose the AQI data at 6:00 am, 9:00 am, 12:00 am 15:00 pm, etc., interval three hours, which are more representative of air pollution emissions.
Point 4: Lastly, You authors would be better explain the algorithms more specifically.
Response 4: this paper used GARCH-BEKK model and motif algorithm of complex network to analyze the air pollution’s spillover. So we added the explain of GARCH-BEKK model on page 3, and the explain of motif model on page 5:
The explain of GARCH-BEKK MODEL:
GARCH-BEKK model often used to analyze the synchronization between time series. Taking this paper as an example: if the AQI in city A rise, then AQI in city B also rise, the AQI in these two cities have
The explain of motif:
The motif is a kind of structure which size between node and modularity in complex network, it’s the basic topologic of complex network. Motif is the basic connect method of nodes in complex network.
Thank you very much for your comments on our article. These comments not only improve the paper but also are of great help to our future scientific research. Thanks again.
The specific modification is in the attachment; please confirm. Thank you very much.
Best regards,
Baocheng Yu

Reviewer 3 Report
Comments:
- I cannot see the legends of Figure 2 (low quality). Please improve its resolution.
- Add few sentences regarding health effects of air pollution In introduction, the authors can use the following paper in Introduction :
- Exposure to high levels of PM5 and PM10 in the metropolis of Tehran and the associated health risks during 2016–2017
- Particulate matters and bioaerosols during Middle East dust storms events in Ilam, Iran
Author Response
Dear Editor and Reviewers:
Thank you for your letter and comments concerning our manuscript entitled “Spatial and seasonal characteristics of air pollution spillover in China” (sustainability-1390246). The comments were all valuable and very helpful for us in revising and improving our paper, and they provided important significant guidance to our research. We have studied the comments carefully and made corrections that we hope meet with your approval. The main corrections in the paper and the response to the reviewers’ comments are below. The revisions are marked in red.
Point 1: I cannot see the legends of Figure 2 (low quality). Please improve its resolution.
Response 1: we improved the resolution of legend in figure 2 on page 6
Point 2: Add few sentences regarding health effects of air pollution In introduction, the authors can use the following paper in Introduction :
Exposure to high levels of PM5 and PM10 in the metropolis of Tehran and the associated health risks during 2016–2017
Particulate matters and bioaerosols during Middle East dust storms events in Ilam, Iran
Response 2: we added these two papers in the introduction on page 1 as literature 43 and 44.
We added the literature citation in the manuscript as: ” Like increasing the cancer risk of cities’ residents, and increasing the number of bacterial colonies in the air, etc. [43-44].”
Thank you very much for your comments on our article. These comments not only improve the paper but also are of great help to our future scientific research. Thanks again.
The specific modification is in the attachment; please confirm. Thank you very much.
Best regards,
Baocheng Yu
